# Antagonistic Effects of IL-4 on IL-17A-Mediated Enhancement of Epidermal Tight Junction Function

**DOI:** 10.3390/ijms20174070

**Published:** 2019-08-21

**Authors:** Matthew G. Brewer, Takeshi Yoshida, Fiona I. Kuo, Sade Fridy, Lisa A. Beck, Anna De Benedetto

**Affiliations:** 1Department of Dermatology, University of Rochester Medical Center, Rochester, NY 14642, USA; 2Department of Dermatology, College of Medicine University of Florida, Gainesville, FL 32610, USA

**Keywords:** atopic dermatitis, tight junctions, cytokines, claudin, STAT3

## Abstract

Atopic dermatitis (AD) is the most common chronic and relapsing inflammatory skin disease. AD is typically characterized by skewed T helper (Th) 2 inflammation, yet other inflammatory profiles (Th1, Th17, Th22) have been observed in human patients. How cytokines from these different Th subsets impact barrier function in this disease is not well understood. As such, we investigated the impact of the canonical Th17 cytokine, IL-17A, on barrier function and protein composition in primary human keratinocytes and human skin explants. These studies demonstrated that IL-17A enhanced tight junction formation and function in both systems, with a dependence on STAT3 signaling. Importantly, the Th2 cytokine, IL-4 inhibited the barrier-enhancing effect of IL-17A treatment. These observations propose that IL-17A helps to restore skin barrier function, but this action is antagonized by Th2 cytokines. This suggests that restoration of IL-17/IL-4 ratio in the skin of AD patients may improve barrier function and in so doing improve disease severity.

## 1. Introduction

Atopic dermatitis (AD) is a chronic, inflammatory skin disorder characterized by an overactive immune response to a host of environmental allergens and dry/itchy skin. Over the last decade, important discoveries have demonstrated that AD develops in part from genetic, and/or acquired defects in the skin barrier. The overall hypothesis is that skin barrier defects allow resident immune cells to gain access to microbes, allergens, and irritants, resulting in a robust type 2 immune response that can be characterized by the tissue recruitment of T helper (Th) 2 cells. Recently, other Th subtypes have been identified in acute (e.g., Th22/Th17) and chronic (e.g., Th22 and Th1) AD lesions, although the importance of these subsets in AD pathogenesis is still unclear [1,2,3].

Epidermal barrier is mediated by two structural components, the stratum corneum (SC) and tight junctions (TJs). TJs, found in the stratum granulosum, seal intercellular spaces and the ‘tightness’ of this structure constitutes a key physiological and dynamic function of the skin. The essential role that epidermal TJs play in maintaining the skin barrier was demonstrated by the claudin (CLDN) 1 deficient mouse, which suffers from extensive transepidermal water loss and succumbs shortly after birth to severe dehydration [4]. We and others have demonstrated functional defects in epidermal TJs from skin samples of AD subjects, which may in part be due to the reduced expression of two transmembrane TJ components, CLDN1 and CLDN23 [5,6,7]. Kast et al. reported that CLDN1 is the most abundant CLDN family member in the skin and in primary human keratinocytes (PHK), with lower levels of the CLDN family members 4 and 25 occurring, while CLDN 3, 5, 7, 8, 10, and 23 were only present in the skin (but not detected in primary culture models) [8].

Altered TJ protein expression has been observed in several human skin disorders including psoriasis, lichen planus, ichthyosis vulgaris, skin wounds, skin infections and in response to ultraviolet exposure; however, the functional consequences of these changes have not been fully evaluated yet [9]. Additionally, very little is known about what regulates epidermal TJ function and claudin expression in immune-mediated skin disorders. Several T-cell derived cytokines were found in AD skin (e.g., IL-4, IL-13, IL-17A, IL-22, IL-25, or TNF-α) and these cytokines were shown to reduce the expression of several SC barrier proteins such as filaggrin, loricrin, S100A11, and involucrin [10,11,12]. In one study, TNF-α was shown to alter TJ protein expression and localization in a skin explant model, as well as TJ function in cultured keratinocytes [9]. Another study showed contradictory results, with no effect observed with either TNF-α or IL-4 treatment, while IL-17 diminished and IL-22 enhanced TJ function in cultured keratinocytes, as measured by transepithelial electric resistance (TEER) [6]. Importantly, in vivo studies have strongly suggested that IL-17A has barrier promoting properties. This was highlighted by increased intestinal permeability in mice lacking IL-17A as well as spontaneous development of allergic skin inflammation in mice lacking the IL-17RA [13,14]. To begin to clarify the effects that AD related cytokines play in barrier function of the skin, we analyzed how the Th17 cytokine, IL-17A, influences TJ barrier properties in cultured keratinocytes and human skin explants and how Th2 cytokines may affect that response.

## 2. Results

### 2.1. IL-17A Enhances Epidermal TJ Barrier Integrity

PHK were exposed to cytokines throughout the differentiation process, thus modeling the effect they would have on TJ assembly during wound repair. In our submerged culture model, we observed that TJ barrier function, quantified by TEER, increases to a peak of approximately 2500 ohms × cm^2^ around 120 ± 24 h after addition of differentiation media [15]. Notably, in our model we observed that as TEER increases, the immunoreactivity of TJ components changes from a dotted, disrupted pattern to a continuous “chicken wire” membranous staining typically observed in mature TJ of the human epidermis (Appendix A).

When PHK were differentiated in the presence of IL-17A, we observed a dose-dependent enhancement of the TJ barrier function by 72 h (Figure 1). This was demonstrated by a significant enhancement of TEER in IL-17A treated PHK (72 h, *p* ≤ 0.01, *n* = 3–6; Figure 1A) and a reduction in the paracellular fluorescein flux (72 h, *p* ≤ 0.05, *n* = 3; Figure 1B). To further validate our findings, we tested the effect of IL-17A on human epidermal samples in an ex vivo model. We tape stripped (15x) discarded healthy human skin, as a model of mechanical disruption and then isolated the epidermis and monitored epidermal TJ recovery with and without IL-17A stimulation. To do this we used a modified micro-Snapwell™ system, previously developed to study TJ function in intestinal epithelium [16], and adapted in our laboratory for epidermal sheets [17]. We observed a significant reduction in the transepidermal fluorescein flux (*p* ≤ 0.02, *n* = 3; Figure 2C) after IL-17A treatment (24 h), thus confirming our barrier-enhancing findings in submerged PHK. Altogether, these observations demonstrate that IL-17A enhances the development of TJ epidermal barrier function.

### 2.2. IL-4 Inhibits IL-17A-Mediated TJ Barrier Enhancement

In contrast to the IL-17A effects on TEER in our model, IL-4 did not significantly alter TJ integrity of cultured PHK monolayers. Previous studies have demonstrated that Th2 cytokines antagonize IL-17A-induced production of antimicrobial peptides and S100A8 in human keratinocytes [18,19,20]. Therefore, we tested whether Th2 cytokines also inhibit IL-17A barrier-enhancing effects. When PHK were treated with both IL-17A and IL-4 we observed that the enhanced TEER and reduced paracellular fluorescein flux observed in response to IL-17A were completely inhibited by co-treatment with IL-4 (50 ng/mL; respectively *p* = 0.024 and *p* = 0.002, Figure 2A,B). Of note, we observed a slight, but not significant (*p* = 0.08) increase in paracellular fluorescein flux with IL-4 treatment at 72 h (Figure 2B), suggesting this cytokine might have an effect on TJ pore size. We again validated our finding in an ex vivo model, co-treatment with IL-4 was able to block the IL-17A induced increase in permeability flux of fluorescein (Figure 2C). To determine whether IL-4 blocks other known IL-17A mediated effects in our in vitro model we measured the expression of S100A7, a well-known downstream product of IL-17A signaling [21]. S100A7, also known as psoriasin, is a calcium-binding protein with chemotactic and antimicrobial properties that is expressed in AD lesions and even more so in psoriasis skin, a Th1/Th17 driven disease [22]. IL-17A-mediated S100A7 expression in PHK was blocked by co-stimulation with IL-4 (Appendix A).

### 2.3. Inhibition of STAT3 Activation Reduces IL-17A-Induced TEER

It has been suggested that Janus kinases (JAK) and mitogen-activated protein kinases (MAPK) are downstream signaling pathways activated by IL-17A [23]. Therefore, we examined if JAK and MAPK activation are involved in IL-17A mediated actions on epidermal TJ function. A major cytosolic target of JAK signaling is the phosphorylation and nuclear translocation of signal transducer and activator of transcription 3 (STAT3). In our model, we confirmed that IL-17A enhanced STAT3 activation in PHK, as demonstrated by increased STAT3 phosphorylation at amino acid Y705 (Figure 3A), suggesting the importance of the JAK-STAT3 pathway in the barrier-enhancing effect of IL-17A. Using a pan-JAK inhibitor (JAK_Tot_) that blocks all four JAK isoforms (JAK inhibitor I; Calbiochem, 10 μM) a significant decrease in the IL-17A-dependent increase of TEER was observed in our PHK model (IL-17A vs JAK_Tot_ + IL-17A, *p* = 0.025; *n* = 4; Figure 3B). By contrast, no inhibition of IL-17A-enhanced TEER was observed from PHK treated with an upstream MAPK inhibitor (PD98056; data not shown). Treatment with JAK_Tot_ completely blocked both baseline and IL-17A-dependent STAT3 activation (Figure 3A). Notably, we observed that IL-4 reduced STAT3 phosphorylation both at baseline and in response to IL-17A treatment (Figure 3C), suggesting that IL-4′s attenuation of the IL-17A barrier-enhancing effect may be mediated by the JAK-STAT3 pathway.

### 2.4. IL-17A Enhances Claudin-4 Expression

To further investigate the mechanisms responsible for IL-17A mediated barrier enhancement, we quantified the expression of two highly expressed epidermal claudins, CLDN1 and CLDN4, in both submerged PHK and in the epidermal organotypic model. This latter model is three-dimensional and more closely mimics mature human epidermis with a well-differentiated, multilayered structure that includes both a stratum granulosum (where TJs reside) and SC layers. Using this model, we detected increased CLDN4 immunoreactivity in the presence of IL-17A (100 ng/mL; *p* = 0.04) (Figure 4A,B). IL-4 in contrast slightly, but significantly reduced CLDN4 expression compared to the media alone (50 ng/mL; *p* = 0.04). Additionally, IL-17A-induced CLDN4 expression was blocked by IL-4 co-treatment (*p* = 0.009; Figure 4A,B). The expression of CLDN1 was not affected by either cytokine alone, or in combination (Figure 4C). The TJ findings in the submerged PHK culture were in line with those observed in the organotypic model. IL-4 stimulation alone and in combination with IL-17A reduced CLDN4 expression as detected by Western blot analysis while CLDN1 expression was unaffected by cytokine treatment (Appendix A).

## 3. Discussion

There is no doubt that type 2 immunity plays a central role in atopic diseases, including AD. Less clear is the contribution of other cytokines, such as IL-17A, in AD pathogenesis. We have now unveiled a novel role for IL-17A in repairing and/or enhancing TJ epidermal barrier function. In this work, we demonstrated that IL-17A enhances TJ integrity at an early time point in differentiating PHK (Figure 1) and this was associated with enhanced expression of CLDN4 (Figure 4).

Notably, IL-17A effects may be inhibited in AD because the antagonistic effect of the Th2 cytokines in the tissue. Previous studies showed Th2 cytokines can reduce IL-17A expression by Th17 cells, as well as IL-17A-induced production of antimicrobial peptides, S100A7 and S100A8 by human keratinocytes [18,19,20,24]. We have added another example to strengthen the paradigm that it is the relative expression of IL-4 and IL-17A that determines the biological consequences of either of these cytokines (Figure 2). IL-17A’s ability to induce antimicrobial peptides as well as recruit neutrophils strongly suggests that this cytokine plays a crucial protective role at epithelial sites. Clinical trials testing the effect of biological therapeutics targeting the Th1/Th17 pathways have been conducted in AD. Anecdotally, good responses in some AD patients were reported with Ustekinumab, an antibody that blocks Th1/Th23 signaling and thus reduces Th17 activation. However, in two independent, randomized, placebo-controlled, phase II studies Ustekinumab did not demonstrate meaningful efficacy in AD. Another trial with Secukinumab that targets the cytokine, IL-17A, was conducted in AD; however, results are not available yet. This trial will be extremely informative to better understand the role of specific cytokines in AD.

We demonstrated in this study that blocking STAT3 activation via a pan-JAK inhibitor was associated with loss of IL-17A-mediated TJ function enhancement (Figure 3). Remarkably, we observed that the IL-4 inhibitory effect on IL-17A’s ability to enhance barrier was partly mediated by reduced STAT3 activation. Since another IL-17A downstream protein S100A7 (also STAT3 dependent) was inhibited by IL-4 co-stimulation, this observation further supports that IL-4 blocks IL-17A signaling at the level of transcriptional control. Activation of STAT3 has been shown to play a key role in many cellular processes such as cell growth and apoptosis. Indeed, STAT3 knockout mice are not viable [25]. K5-Cre/STAT3 transgenic mice, whose epidermal and follicular keratinocytes lack STAT3, were viable and the development of the epidermis and hair follicles appeared normal. However, hair cycle and wound healing processes were severely compromised [26]. When STAT3 is inactivated in intestinal epithelial cells through lentiviral transduction of a dominant negative STAT3 protein, barrier is severely impacted [27]. STAT3 activation was also shown to increase permeability in human retinal endothelial cells [28]. Contrasting with our data, another group showed that baseline STAT3 activation was not reduced when PHK were treated with IL-4 and -13 together [29]. This could be explained because IL-13 can also induce phosphorylation of STAT3 in PHK [30]. In humans, STAT3 mutations cause a form of the Hyper IgE syndrome (AD-HIES; OMIM: 147060). Severe AD-like phenotype, increased susceptibility to S. aureus infections and eosinophilia are some of the clinical features of this disease, in addition to reduced peripheral Th17 cells (STAT3 is involved in Th17 differentiation cells) [31]. However, involvement of the STAT3 pathway in TJ function and composition in the skin has not been directly evaluated in these patients.

An alternative and complementary, approach to investigate the effect of cytokines on TJ function is to directly look at the expression and localization of key TJ components. In this study, we investigated the effect of the IL-17A with or without IL-4 on the expression of CLDN1 and 4, the two most abundant CLDNs in human keratinocytes [8]. Our studies demonstrated that CLDN4, but not CLDN1 expression is enhanced by IL-17A, and this enhancement is prevented by co-treatment with IL-4 (Figure 4). Previous work by Kirschner et al. showed that silencing CLDN4 in PHK reduced TEER and enhanced permeability to small macromolecules (4 kDa dextran-FITC conjugated). CLDN23 is another TJ-associated molecule we hypothesized to be important in barrier function since it is down-regulated (mRNA) in the skin of AD patients [5]. Currently, the reagents for this protein are poorly validated in human skin, which is why it was not investigated. Future studies will have to clarify the specific effects of reduced expression of each of the claudins in AD subjects

Our findings in an epidermal in vitro model are in contrast with previous data showing IL-17A decreased TEER in ocular epithelial cells, suggesting impaired barrier function capabilities [32]. Gutowska-Owsiak D. et al. reported findings from a microarray study in HaCaT cells showing that IL-17A treatment (20 or 50 ng/mL for 24 h) reduced expression of several intercellular junction components, implying IL-17A could reduce keratinocyte barrier function at the level of the SC, but a link to TJs was not suggested [21]. It is important to point out that studying the mechanisms regulating epithelial TJ function and composition is made difficult by pleiotropic cytokine actions, in addition to temporal and dose-dependent variation in cell culture systems [33]. Additionally, the study of epidermal TJs is further complicated by different culture conditions employed, which also typically results in variable levels of differentiation. To minimize many of these sources of variability we chose to focus our studies on submerged keratinocyte cultures because they form robust barrier at the level of the TJ (Appendix A), which we believe more faithfully mimics the epithelium. Air liquid cultures cornify which is ideal for studying the SC, but importantly they do not form robust TJ, making this an unreliable model to study the effects of cytokines on barrier formation. This biological complexity might explain the discrepancy of data published so far in this field and as such requires further validation in vivo.

Dupilumab (Regeneron, Tarrytown, NY, USA & Sanofi, Paris, France) a fully human monoclonal antibody against the Th2 cytokine receptor IL-4Rα, is the first biologic medication approved worldwide for the treatment of AD. This has been a revolutionary treatment option for patients with AD, with a good safety profile [34]. This fully humanized monoclonal antibody targets the common receptor subunit (IL-4Rα) used by both IL-4 and IL-13 and thereby blocks signaling. We hypothesize that some of the clinical improvement and normalization of the epithelial transcriptome may be due to diminished antagonism of IL-17A. A better understanding of the complex interactions between relevant Th cytokines found in AD lesions and how they affect epidermal barrier function will be critical for a more thorough understanding of AD pathogenesis. The development and study of highly targeted AD therapies will likely provide an opportunity to look at this issue and allow us to tailor therapies to individual patient’s tissue inflammation.

## 4. Methods

### 4.1. Isolation and Culture of Primary Human Foreskin Keratinocytes

Human keratinocytes were isolated from neonatal foreskin as previously described [35]. PHK were cultured in Keratinocyte-SFM (Thermo Fisher Scientific, Waltham, MA, USA) with 1% Pen/Strep, 0.2% Amphotericin B (Thermo Fisher Scientific, Waltham, MA, USA). To differentiate PHK, cells were grown in DMEM (11965092, Thermo Fisher Scientific, Waltham, MA, USA) with 1% Pen/Strep, 0.2% Amphotericin B. For cytokine experiments, the following reagents were added alone or in combination to the culture media from the time of differentiation and replaced with every 48-h media change: human IL-4 (5–50 ng/mL; R&D system, Minneapolis, MN, USA), human IL-13 (5–50 ng/mL; R&D system, Minneapolis, MN, USA), human IL-17A (1–100 ng/mL; R&D system), JAK inhibitor I (10 μM; Calbiochem, San Diego, CA, USA), and PD98056 (10 μM; Calbiochem, San Diego, CA, USA). For the epidermal organotypic model experiment, keratinocytes were grown as previously described [36]. For all cytokine treatment studies, keratinocytes were starved of growth factors for 24 h before stimulation with the indicated cytokines.

### 4.2. Transepithelial Electric Resistance (TEER) and Paracellular Flux

TJ integrity in cultured PHK was evaluated using two complementary assays, TEER and paracellular flux, as previously described [5]. Briefly, TEER was measured using an EVOMX voltohmmeter (World Precision Instruments, Sarasota, FL, USA). To evaluate the paracellular flux across PHK cultures, 0.02% Fluorescein (FITC) Sodium (Sigma, St. Louis, MO, USA) in HBSS was added to the upper chamber. Samples were collected from the lower chamber at different times (0.5–3 h). Paracellular permeability was presented as “Relative Fluorescence Flux” = experimental condition/media alone x 100.

We used the Micro-Snapwell™ barrier function assay (Corning Incorporated, Corning, NY, USA) to determine the permeability of ex vivo human skin. The protocol was adapted from previously described work [16]. Full thickness epidermis was isolated from discarded, deidentified skin samples. To disrupt skin barrier, the epidermis was taped stripped 15 times and then removed using a Weck blade (Goulian Skin Graft Knife Set, George Tiemann & Co. Hauppauge, NY, USA) and washed in PBS. Skin samples were mounted on filter supports (Whatman Nuclepore Track-Etch Membrane; GE Healthcare Biosciences, Pittsburgh, PA, USA). The epidermal side was faced upward and sandwiched between two custom made Plexiglas discs containing an opening of 3 mm. This sandwich was then placed in the modified Snapwell™ chambers. Samples were immersed in DMEM media and kept at 37 °C, 5% CO_2_ for 30 min. Media was added to samples and to test the effect of IL-17A, 50 ng/mL of the cytokine was incorporated into the media, which was then added to both sides of the transwell. Paracellular flux of fluorescein into the lower levels of human epidermis was measured at 24 h later as described above. The Pathology Department at the University of Rochester Medical Center provided the discarded human skin, which was approved by the institutional Research Subject Review Board (RSRB 00042616; approved on 31 May 2012).

### 4.3. Immunoblotting

PHK were placed on ice in RIPA lysis buffer (20 mM Tris, 50mM NaCl, 2mM EDTA, 2mM EGTA, 1% sodium deoxycholate, 1% TX-100, and 2% SDS, pH 7.4) with a 100-fold dilution of both Protease Inhibitor Cocktail (Sigma, St. Louis, MO, USA) and Phosphatase Inhibitor Cocktail (Sigma, St. Louis, MO, USA) for 30 min. The samples were boiled for 10 min and then centrifuged for 15 min. Forty µg of protein, as determined by the BCA (Thermo Fisher Scientific, Waltham, MA, USA) protein quantification assay, were mixed in NuPage LDS Sample Buffer (Thermo Fisher Scientific, Waltham, MA, USA) and applied to 4-12% NuPage Bis-Tris gels (Thermo Fisher Scientific, Waltham, MA, USA). Electrophoresis was performed under reducing conditions with MES SDS Buffer (Thermo Fisher Scientific, Waltham, MA, USA). Membranes were incubated in blocking solution, which contained 5% non-fat dry milk + 0.05% Tween-20 in PBS, for 1 h at RT and then incubated with primary antibodies: claudin-4 (ab53156; Abcam, Cambridge, UK), claudin-1 (JAY.8; Thermo Fisher Scientific, Waltham, MA, USA) STAT3 (STAAD22A; Abcam, Cambridge, UK), pSTAT3 (EP2147Y, phospho-Y705; Abcam, Cambridge, UK), S100A7 (47c1068; Thermo Fisher Scientific, Waltham, MA, USA) and β-Actin (C4; Santa Cruz, Dallas, TX, USA). HRP-linked secondary antibody (GE Healthcare Biosciences, Pittsburgh, PA, USA) was used to visualize protein staining in tandem with ECL reagent (GE Healthcare Biosciences, Pittsburgh, PA, USA) by autoradiography with Kodak BioMax MR film (Kodak, Rochester, NY, USA).

### 4.4. Immunostaining

PHK were grown on a cover glass were fixed in methanol at −20 °C for 15 min, followed by blocking with 1% BSA in PBS and immunolabeled with the above pSTAT3 antibody for 2.5 h at 4 °C. This was followed by a 1-h incubation with secondary antibodies. A 1:1000 dilution of Alexa Fluor 568 donkey-anti-mouse IgG H+L (Thermo Fisher Scientific, Waltham, MA, USA) and 1:10,000 4’,6-diamidino-2-phenyl-indole, dihydrochloride (DAPI) (Molecular Probe, Eugene, OR, USA) was used. Fluorescent images were taken using an Olympus FV1000 laser scanning confocal microscope with the FV10-ASW 1.7 software (Olympus, Tokyo, Japan). The saturation level of fluorescence intensity was determined for the controls using the Hi-Lo feature of the Fluoview software. In each experiment, the images captured used identical settings during acquisition and no further manipulation of the images occurred before into importing into the FV1000 software. The organotypic samples were fixed in formalin then embedded in paraffin. Five µm sections were rehydrated after deparaffinization. Slides were incubated in EDTA containing solution, pH 8.0 at 95 °C for 10 min. Samples were incubated overnight at with the same primary Abs used for immunoblotting at 4 °C. After primary antibody binding a 1-h incubation with the secondary antibody Alexa Fluor 488 donkey-anti-rabbit IgG H+L (Thermo Fisher Scientific, Waltham, MA, USA) diluted 1000-fold was done. Fluorescent images were obtained with an Olympus B× 60 microscope

### 4.5. Statistical Analysis

Statistical significance was tested using either a pairwise analysis T test (Mann-Whitney) or the one-way analysis of variance (Kruskal-Wallis) to determine significance. Statistical analysis was accomplished using GraphPad Prism (GraphPad Software, Inc., San Diego, CA, USA). A value of *p* < 0.05 qualified as significant.

## Figures and Tables

**Figure 1 ijms-20-04070-f001:**
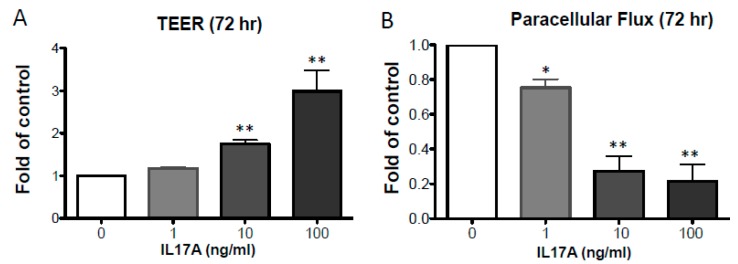
IL-17A enhances epidermal TJ barrier integrity. In PHK, IL-17A dose-dependently (**A**) enhanced TEER (*n* = 3–6) and (**B**) reduced paracellular flux of fluorescein (*n* = 3) 72 h after cytokine treatment. Data are shown as mean ± SEM fold of control. Significance was calculated compared to untreated controls. * *p* ≤ 0.05, ** *p* ≤ 0.05.

**Figure 2 ijms-20-04070-f002:**
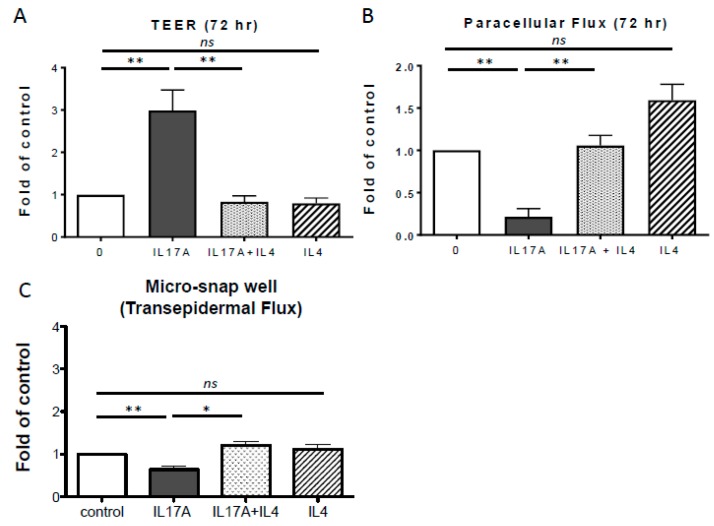
IL-4 inhibits IL-17A mediated barrier enhancement in PHK. Co-treatment with IL-4 (50 ng/mL) (**A**) inhibited IL-17A (100 ng/mL) increased TEER and (**B**) enhanced paracellular flux (*n* = 3–5). (**C**) IL-17A treatment reduced transepidermal flux of fluorescein (50 ng/mL 0.02% fluorescein; *n* = 3) in tape-stripped human skin samples. Data are shown as mean ± SEM fold of control. * *p* ≤ 0.05, ** *p* ≤ 0.01, ns: not significant.

**Figure 3 ijms-20-04070-f003:**
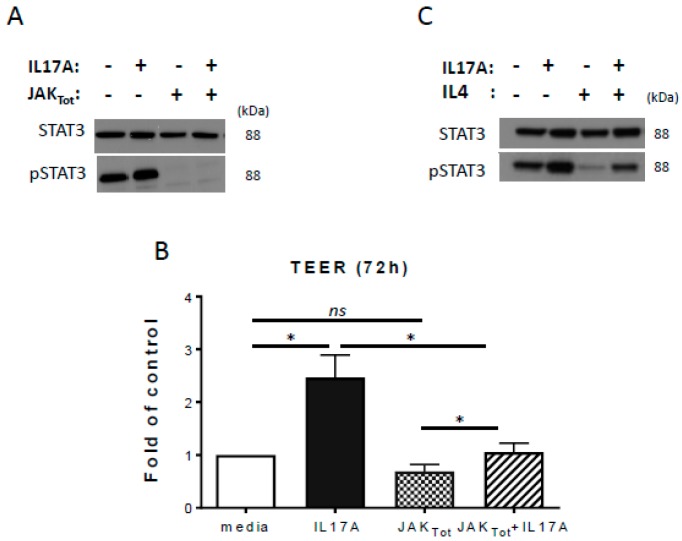
A pan-JAK inhibitor blocks IL-17A-induced TEER enhancement. (**A**) A representative Western blot demonstrates that pan-JAK inhibition blocks STAT3 activation (pSTAT3; EP2147Y), but has no effect on total STAT3 expression (STAAD22A) (**B**) Pretreatment with a pan-JAK inhibitor (JAKtot: 10 μM; *n* = 4) inhibited IL-17A (100 ng/mL) induced TEER in primary human keratinocytes. (**C**) IL-4 blocked STAT3 phosphorylation in PHK treated with IL-17A or media alone. Representative Western blot bands of *n* = 3 experiments are shown. All data are shown as mean ± SEM fold of control. * *p* ≤ 0.05, ns: not significant.

**Figure 4 ijms-20-04070-f004:**
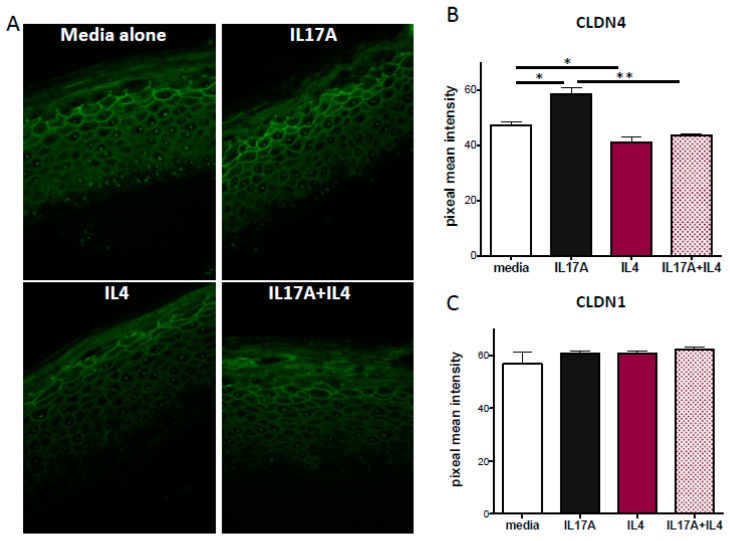
IL-17A increases CLDN4, but not CLDN1 in an organotypic skin model. IL-17A (100 ng/mL), IL-4 (50 ng/mL) or a combination of the two were added to the media of organotypic cultures for 72 h after 9 days of growth. (**A**) Paraffin-fixed sections were stained with either a CLDN4 (pAb) or CLDN1 (JAY.8; not shown) antibody and detected with an Alexa Fluor 488-conjugated secondary (green). (**B**) CLDN4 and (**C**) CLDN1 immunostaining intensity analysis. Images were acquired at the same instrumental setting and at 40× magnification. Mean pixel intensity (5 random fields) was measured using ImageJ. All data are shown as mean ± SEM. * *p* ≤ 0.05, ** *p* ≤ 0.01.

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
