# Peer review of "Antagonistic Effects of IL-4 on IL-17A-Mediated Enhancement of Epidermal Tight Junction Function"

_ijms, 2019, doi:10.3390/ijms20174070_

Round 1

Reviewer 1 Report

The paper describes the investigation of the impact of the canonical Th17 cytokine, IL-17A, on barrier function and protein composition in primary human keratinocytes and human skin explants. The studies demonstrated that IL-17A enhanced tight junction formation and function in both systems, with a
dependence on STAT3 signaling. 

The authors have conducted a sound scientific work and produced a well-written manuscript. The experimental results have been properly recorded and interpreted and the discussions took into consideration the current available literature. Conclusions are supported by the experimental results. Therefore, it is my opinion that the article is suitable for publication in its current form.

Author Response

Thank you for your kind comments.

Reviewer 2 Report

The manuscript of Brewer  et al. named ”Antagonistic Effects of IL-4 on IL-17A-Mediated 3 Enhancement of Epidermal Tight Junction Function” are dedicated to the important problem of atopic dermatitis (AD) related to severely compromised barrier function of the skin developing during this condition. Abnormal skin in AD permit microbes and other alien factors elicit anomalous immune response with  itching and lesions. Authors investigated the impact of the IL-17A cytokine on barrier function in primary human keratinocytes and human skin explants, showing gradual enchantment of barrier function by IL-17A mediated by JAK and affecting claudin expression. Authors have shown that restoration of IL-17/IL-4 ratio in the skin of AD patients may improve barrier function.  This data is really important and interesting and confirm previous data received on claudine deficient animals. Manuscript is clearly  written in good English.

This reviewer can suggest only a minor corrections, as follows:

Line 65-66: 2,000 ohms x cm2 around 96 24 hrs after addition of – It is not clear how many hours, please, correct

Line 99-100: the expression of S100A7 (psoriasin), a well known IL-17A induced downstream gene – psoriasin is product of the gene induced by IL-17A, not the gene itself. Please, correct

Author Response

Thank you for your review; we have made the corrections as suggested in the revised manuscript.